# Preparation and Reflectance Spectrum Modulation of Cr_2_O_3_ Green Pigment by Solution Combustion Synthesis

**DOI:** 10.3390/ma13071540

**Published:** 2020-03-27

**Authors:** Shuhe Ai, Huade Zheng, Jincheng Yu

**Affiliations:** 1School of Materials Science and Engineering, South China University of Technology, Guangzhou 510006, China; shai@scici.cn; 2South China Institute of Collaborative Innovation, Dongguan 523808, China; 3National Engineering Research Center for Tissue Restoration and Reconstruction, South China University, of Technology, Guangzhou 510006, China; 4Key Laboratory of Biomedical Engineering of Guangdong Province, and Innovation Center for Tissue, Restoration and Reconstruction, South China University of Technology, Guangzhou 510006, China; 5School of Materials Science and Engineering, Shandong University, Jinan 250061, China; Jincheng.yu@manchester.ac.uk

**Keywords:** chromic oxide green pigment, submicron Cr_2_O_3_ crystal, solution combustion synthesis, diffuse reflectance spectrum, green plants

## Abstract

An amorphous precursor of Cr_2_O_3_ pigment was prepared via solution combustion synthesis. After calcination at 1000 °C for 1.0 h, the precursor was converted into well-crystallized submicron Cr_2_O_3_ crystals with uniform particle distribution and low aggregation. Furthermore, Ti, Co, and Fe were doped into the lattice of Cr_2_O_3_, and the effects of these dopants on the reflectance spectroscopy modulation as well as chromatic properties were investigated in detail. As a result, a series of Cr_2_O_3_ pigment samples sharing similar spectra within the wavelength from 400 to 1600 nm with green plants could be successfully fabricated.

## 1. Introduction

In recent years, chromic oxide (Cr_2_O_3_) green pigment has been extensively utilized due to its high stability, pronounced tinting strength, good migration resistance, and low cost [1,2,3]. Notably, sharing the similar color with green plants, it has been used to manufacture camouflage coatings for a long time [4]. However, through an analysis of diffuse reflectance spectra, it can be confirmed that the spectral reflectance characteristics of chromic oxide green pigment is apparently different from that of natural plants.

Figure 1 shows the approximate reflectance spectrum of green plants. For most natural plants, their intrinsic green colors are ascribed to the characteristic reflection peaks at 550 nm on the reflectance spectrum. The peak values highly depend on the plant species, the leaf age, and the chlorophyll, with fluctuations of 10% to 20% even more. Due to the characteristic absorption of chlorophyll, two valleys (around 450 nm and 680 nm) are distinguished on visible bands of the reflectance spectrum. By marked contrast, an upsurge of reflectance can be found when the wavelength increases from 680 to 800 nm, after which the reflectance keeps at a high level until the wavelength reaches 1300 nm. Therefore, it can be confirmed that a platform of near-infrared light (NIR) is formed. In reality, the average reflectance of NIR platform varies from 40%–70% according to the species of plants. Besides, moisture in leaves can selectively absorb light in the region of 1300–1600 nm, thereby a valley is revealed in the corresponding area of the diffuse reflectance spectrum [5]. Compared to natural plants, the average reflectance of Cr_2_O_3_ pigment in the region of 800–1300 nm is relatively lower while the highest reflection peak can reach over 30%. Furthermore, no valley can be found during 1300–1600 nm due to the absence of moisture. All these differences make it easy to distinguish traditional Cr_2_O_3_ green camouflage coatings with green plants by means of modern detection equipment.

Recently, it has been reported that the reflectance spectrum of Cr_2_O_3_ can be modulated by doping Al, Mo, La, Pr, V, Ti, Fe, et al. [6,7,8]. Effects of correlative factors, including the particle size, uniformity, crystal boundaries, and crystal defects on NIR reflectance are studied at the same time. In summary, crystal boundaries and defects prove to be non-negligible factors that could absorb near infrared light and decrease NIR reflectance [8]. Particle size and distribution affect diffuse reflection following the Fresnel formula, since the particle sizes of most samples prepared in these work are of the same order of magnitude with a wavelength of visual near-infrared light [9].

However, most previous studies mainly focused on promoting the NIR reflectance of Cr_2_O_3_ in order to provide a highly NIR reflective cool pigment. The comprehensive influence of these dopants on visible NIR diffuse reflectance spectra was neglected. Besides, the difficulties in controlling particle size as well as in the elimination of crystal defects reduced the experimental accuracy of previous studies.

In this study, an ideal chromic oxide green pigment consisting of submicron Cr_2_O_3_ crystals was introduced at first. To be more precise, the submicron Cr_2_O_3_ crystals are expected to be separated and fully crystallized with low aggregation. Under such conditions, adverse effects of the crystal boundary or defects on NIR reflectance could be ignored. Therefore, the preparation of submicron Cr_2_O_3_ crystals could offer a foundation for the further research of reflectance spectrum modulation. After that, Cr_2_O_3_ pigments were prepared via an improved solution combustion synthesis method [10,11]. Ti, Co, and Fe were doped into the lattice of Cr_2_O_3_, and the effects of these dopants on reflectance spectroscopy modulation as well as chromatic properties were investigated. Finally, a series of Cr_2_O_3_ pigments sharing similar spectra within the wavelength from 400 to 1600 nm with green plants were prepared.

## 2. Materials and Methods

Pure Cr_2_O_3_ and Ti/Co/Fe doped Cr_2_O_3_ green pigments were prepared via solution combustion synthesis, respectively. Cr(NO_3_)_3_·9H_2_O (chromium nitrate, 99.5%), Co(NO_3_)_2_·6H_2_O (cobaltous nitrate, 99.0%), Fe(NO_3_)_3_·9H_2_O (ferric nitrate, 99.0%), C_6_H_8_O_7_·H_2_O (citrate acid, 99.5%), CO(NH_2_)_2_·4H_2_O (urea, 99.5%), PEG 200 (polyethylene glycol, 99%), C_16_H_36_O_4_Ti (tetrabutyl titanate, 99%), and deionized water were used as starting materials.

Taking synthesizing 0.1 mol Cr_2_O_3_ as an example, 80 mL of deionized water was heated to 60 °C and then, the initial solution was acquired after 7.0 g of citrate acid and 10 g of urea were dissolved. A certain amount of tetrabutyl titanate was dispersed in 12.0 g of PEG 200, and the mixture was added to the above solution by individual drops. Different amounts of Co(NO_3_)_2_·6H_2_O, Fe(NO_3_)_3_·9H_2_O, and 0.2 mol Cr(NO_3_)_3_·9H_2_O were dissolved in the solution in order. The molar ratios of Cr, Ti, Co, and Fe for each sample are listed in Table 1; thus, the amount of tetrabutyl titanate, cobaltous nitrate, and ferric nitrate that is needed can be figured out. During the whole process, stirring was in need to promote dissolution, and the heating temperature was kept at 60 °C. Then, the mixed solution was evaporated and concentrated until half of the volume was left.

Solution combustion synthesis was conducted with the aid of a self-designed device called a self-propagating combustion furnace. Figure 2a shows the diagrammatic sketch of this furnace, where the flame nozzle performs as the core component. Equipped with a corundum tube that is highlighted in the schematic, the details of the flame nozzle are exactly displayed in Figure 2b. Cr_2_O_3_ powders were synthesized as follows. Firstly, the corundum tube was heated to 500 ± 1 °C and then concentrated solution stored in the reservoir was pumped into the silicon tube with an constant velocity (60–120 mL/min) using a peristaltic pump. A quartz tube inserted in the hole of a corundum plug acted as the bridge between the silicon tube and corundum plug to protect the silicon tube from thermal damage. No praying equipment was involved in this research. Once the solution entered the tube, continuous combustion synthesis was ignited due to the high inflammability of the mixture of nitrates and organics. With the use of flame nozzle, powders generated in the flame would be pushed out of the corundum tube (shown in Figure 2b) immediately, and then the combustion synthesis process could be continued without blocking issues. The powders collected in the recovery tank are described as the precursors of Cr_2_O_3_ because subsequent heat treatment is still required to ensure the complete reaction, thereby improving the crystallinity. In this research, the precursor of S1 was divided into two batches and then calcined at 900 °C and 1000 °C for 1.0 h respectively with a heating rate of 10 °C/min. The calcination temperatures of remaining samples were fixed at 1000 °C.

The phase identification of the precursor and Cr_2_O_3_ samples were carried out by powder X-ray diffraction (XRD, Rigaku, RINT-2000, Osaka, Japan) using Cu-Kα radiation. The morphology of samples was investigated using a scanning electron microscopy (Merlin, Carl Zeiss, Oberkochen, Germany). The visible NIR diffuse reflectance spectra (400–2500 nm) of Cr_2_O_3_ samples were measured by a UV-vis-NIR spectrophotometer (LAMBDA750, PerkinElmer, Waltham, MA, USA). The diffuse reflectance spectra (400–700 nm) and colorimetric values (reported in a CIEL*a*b* colorimetric system) of Cr_2_O_3_ samples as well as natural plant leaves were measured on an automatic differential colorimeter (SC800, CHN Spec, Hangzhou, China). All the optical photographs presented in this paper were taken by a digital camera (D5600, Nikon, Tokyo, Japan).

## 3. Results and Discussion

### 3.1. Powder X-Ray Diffraction Analysis

The XRD patterns of Cr_2_O_3_ samples calcined at 900/1000 °C and the precursor without calcination are depicted in Figure 3. The absence of peaks in the precursor indicates a typical amorphous structure, which is different from the results in similar studies of solution combustion synthesis [10,11]. In this work, the additions of citrate acid, urea, and PEG 200 were optimized by multiple pre-experiments, and then stable combustion at lower temperatures was created. With the utilization of a flame nozzle in the combustion furnace, the precursor generated can merely stay in the high-temperature area of the flame for seconds before it is pushed out of the nozzle. The lower temperature of combustion and the short time exposed to the flame cannot guarantee the crystallization of Cr_2_O_3_; thus, the amorphous structure is kept in the precursor.

After calcination at 900/1000 °C for 1.0 h, XRD patterns of the undoped Cr_2_O_3_ samples (S1-900, S1-1000) are in precise agreement with Cr_2_O_3_ (eskolaite, PDF reference pattern: 01-072-1207). Figure 3 also illustrates the XRD patterns of the doped Cr_2_O_3_ samples (S3, S6, S12) calcined at 1000 °C, where all the prominent diffraction peaks were indexed as Cr_2_O_3_ (eskolaite, PDF reference pattern: 01-072-1207), indicating that the addition of small amounts of Ti/Co/Fe does not significantly change the phase composition of Cr_2_O_3_. Besides, an emphasized view on the patterns from 32.5 to 37.5 degrees displayed a distinct shift of the peak toward a lower angle after Ti/Co/Fe was doped. From the periodic table of elements, it is easy to know that the atomic radius of Ti, Cr, Fe, and Co decreases sequentially. The same shift trend of diffraction peaks for the S3, S6, and S12 samples is probably because 4 mol% Ti among the dopants plays a major role during the doping process. Based on the Bragg equation, it could be inferred that the interplanar space increased when Cr was substituted by Ti, resulting in the left shift of diffraction peaks.

### 3.2. Morphological Analysis

Figure 4 gives information about the morphology of both precursors and calcined powders. Figure 4a shows that the precursor has an amorphous structure, which is well matched with the XRD result. A series of pores are distributed randomly, and a plate-like structure is more likely to be formed at this stage. Figure 4b, c displays the morphology of S1 calcined at different temperatures. It can be found that well-defined sub-micron crystalline grains with slight aggregation are observed in both two samples, while the grain size of S1-1000 is larger than that of S1-900, indicating that a higher calcination temperature is beneficial to accelerating the growth of Cr_2_O_3_ grains. When compared with Cr_2_O_3_ samples prepared by a hydrogen reduction of chromite ore [1] and thermal decomposition of chromium hydroxide [12,13], the serious aggregation and grain size distribution are efficiently optimized. From Figure 4d, we can know that the morphology of S12 is similar to that of S1-1000, where the grain size is estimated to be about 0.4 μm, implying that Fe/Co/Ti doping has no negative effects during this process. Therefore, the solution combustion synthesis is approved to be a good approach to obtain submicron Cr_2_O_3_ crystals with better crystallinity and uniform distribution.

### 3.3. Visible Near-Infrared Diffuse Reflectance Spectra Analysis

The diffuse reflectance spectra of Cr_2_O_3_ samples calcined at various temperatures and samples with Ti/Co/Fe doping have been measured, and the analysis results are shown as below. According to Figure 5, in the NIR range (780–2500 nm) of the pure Cr_2_O_3_ sample, the reflectance of S1-1000 is larger than that of S1-900, while there is no significant difference between the two spectra in the range from 400 to 780 nm. It can be concluded that calcination at 1000 °C creates a better condition for improving the NIR reflectance of Cr_2_O_3_ samples. Therefore, the calcination temperatures of the remaining samples (S2 to S12) are fixed at 1000 °C. Compared with S1, significantly larger NIR reflectance can be achieved in Ti-doped Cr_2_O_3_ samples S2 and S3, which show promising potential to be used as cool pigments. Meanwhile, a small rise of reflection peak at about 540 nm can be found after Ti doping. Although reflectance in the region of 800–1300 nm is improved, diffuse reflectance spectra of Ti-doped Cr_2_O_3_ is still different from that of green plants. Further work is needed to lower the reflection peaks at around 550 nm and simulate the valley in the region of 1300–1500 nm caused by moisture absorption.

As shown in Figure 6, NIR reflection in the range of 1200–1700 nm for Co-containing Cr_2_O_3_ samples (S4) is significantly lower than that of S1-1000. This change can be attributed to the characteristic absorption of Co^2+^ according to previous studies [8]. Although a characteristic absorption of Co^2+^ (1200–1700 nm) cannot perfectly match with the first valley (1300–1600 nm) on the spectra of green plants, the gap of reflectance between Cr_2_O_3_ pigments and green plants is significantly narrowed, which is beneficial to reducing their recognition. Indeed, Co has already been used to produce Cr_2_O_3_ pigments used for camouflage by the Shepherd color company [14], as an absorption band of Co^2+^ is still the best choice to simulate the valley (1300–1600 nm) of spectra of green plants until now. Besides, Co doping decreases the reflection peak at 540 nm from 30% (S1-1000) to 22% (S4). For samples S5 and S6, in which Co content is fixed at 4 at.%, the NIR reflection can still be improved after Ti doping when compared with S4, and the overall shapes of absorption bands are quite similar. Herein, it makes sense that Ti and Co can modulate the spectra of Cr_2_O_3_ when the amounts of additives are tiny. This is probably because Ti and Co are uniformly doped into the Cr_2_O_3_ crystal lattice rather than forming grain boundary segregation and secondary phases. Furthermore, NIR platforms emerge on the reflectance spectra of Co-containing Cr_2_O_3_ samples (S4, S5, S6), and the height of the platform can be adjusted flexibly by changing the amount of Ti.

The spectra of Fe and Ti co-doped samples (S8, S9, S10) are shown in Figure 7. It is clear that the reflection peak at around 550 nm decreases gradually with the addition of Fe. Although Fe doping reduces the NIR reflectance in the range of 800–1300 nm, it is tolerable considering that the reflectance is still higher than that of most green plants. Therefore, Fe can be introduced into Ti and Co co-doped samples to make further efforts to lower the reflection peak at around 550 nm.

As Figure 8 shows, the reflectance at around 550 nm of Ti, Co, and Fe co-doped samples is between 10% and 20%, so it matches with most green plants. Each spectrum of these samples owns an apparent NIR platform and an absorption band of Co^2+^. Particularly, the reflection spectra (400–1600 nm) of S10, S11, and S12 are in good accordance with that of the green plants given in Figure 1. In conclusion, our doping strategy is convinced to take effects while fabricating camouflage coatings.

Unfortunately, the diffuse spectra analysis shows that all the samples prepared fail to create the same characteristics of the waveband from 1600 to 2500 nm on the reflectance spectrum of green plants. In this research, no helpful element is found to simulate water absorption features at 1900 nm or 2500 nm. However, these pigments still have potential utilization in camouflage coatings for some objective reasons, which are as follows. First, only a small portion of lights in the region of 1600–2500 nm can reach the ground due to the radiation characteristics of solar and moisture absorption of the atmospheric layer [15]. Moisture absorption occurs again before the lights reflected by green plants and camouflage coatings are detected by the visible light and near-infrared sensors. So, it becomes extremely difficult to make a distinction between natural green plants and artificial camouflage coatings when the analysis is conducted on a waveband from 1600 to 2500 nm. In this case, the waveband from 1600 to 2500 nm is rarely used as an operation band by most of the army’s detection equipment. Similar studies also feature discussions on this waveband [16,17]. All in all, the defects of Ti, Co, and Fe co-doped samples can be acceptable for practical applications. However, the authors still believe that some elements not mentioned in this article may be useful to create a perfect match, and many more studies still need to be done via the solution combustion synthesis method that we improved.

### 3.4. Chromatic Properties Analysis

Further study is conducted to evaluate the chromatic properties of some samples to make sure if they can be used to simulate the color of natural green plants. Ficus microcarpa, a kind of widely distributed evergreen tree of southern China as well as South and Southeast Asia, is chosen as a representative of natural green plants for its changeable green color. Diffuse reflectance spectra in the range of visible light (400–700 nm) of these samples as well as several Ficus microcarpa leaves (A to F) are shown in Figure 9. Meanwhile, the similar color to every sample and leaf generated by the colorimeter are listed after the serial number. Figure 9 shows that every Ficus microcarpa leaf tested has a reflection peak around 550 nm and the peak value ranges from 8% to about 25%. In contrast, S1 and S3 have reflection peaks around 540 nm, and their peak values are more than 30%. Obviously, the doping of Co lowers peak values, and then the spectra of S6 is close to the spectra of leaf A and B. As for Ti, Co, and Fe co-doped samples (S10, S11, S120), reflection peaks are getting even lower and the peak values range from nearly 16% to about 10%. Meanwhile, the positions of the reflection peaks shift from 540 nm (S1, S3, S6) to 550 nm (S10, S11) and 560 nm (S12) as the addition of Fe increases. Figure 9 also shows a reflection peak around 410 nm, which is noticeable on the spectrum of pure Cr_2_O_3_ sample (S1), which is gradually smoothed out by the doping of Ti/Co/Fe. It can be found the spectra of sample S10 and leaf D are a good match in the range of 400–600 nm. Generally, samples S6, S10, S11, and S12 all have the potential to make green camouflage coatings considering that the average diffuse reflectance at 550 nm of most green plants is between 10% and 20%. However, it also needs to be pointed out that the doping of Ti/Co/Fe fails to simulate the valley around 680 nm on the reflectance spectrum of green plants, and further work still need to be done.

The L*, a*, and b* values of Cr_2_O_3_ samples as well as Ficus microcarpa leaves (A–F) are listed in Table 2. For Ficus microcarpa leaves, the values of L* and b* decrease quickly, but the value of a* changes little as the color turns from yellowish-green to deep-green. It can be found the value of a* of the undoped Cr_2_O_3_ sample S1 is much lower than those of the Ficus microcarpa leaves tested in this study, and Co, Fe doping is effective to improve the value of a*. So, compared with the undoped Cr_2_O_3_ sample (S1), the Ti, Co co-doped sample (S6) and Ti, Co, Fe co-doped sample (S11) have values of a* that are much closer to those of the leaves. Besides, we also find that the values of a* of S11 and S12 are already too high, indicating that the addition of Fe needs to be cut down when further study is conducted. The color variation of doped Cr_2_O_3_ samples can also be found in the photographs of Cr_2_O_3_ tablets prepared for the color test shown in Figure 10.

## 4. Conclusions

Cr_2_O_3_ green pigments with a submicron scale, fine crystallization and low aggregation were successfully prepared by solution combustion synthesis. The study also proves that Ti, Co, and Fe all have an obvious effect on modulating the reflectance spectra of Cr_2_O_3_ pigments. Doping Ti can significantly improve the NIR reflection of Cr_2_O_3_ as well as improve the reflection peak around 540 nm slightly. The characteristic absorption band of Co^2+^ can be found in the range of 1200–1700 nm after doping Co, and this band can be used to simulate the moisture absorption on the spectra of natural green plants. The addition of Fe can lower the reflection peak around 550 nm while keeping the NIR reflection at a high level. A series of Cr_2_O_3_ pigments with reflection peaks ranging from 12% to 22%, on the NIR platform and an absorption band of Co^2+^ were fabricated by Ti, Co and Ti, and Co, Fe co-doping. These pigments share similar diffuse reflectance spectra with natural green plants within the wavelength from 400 to 1600 nm. The chromatic coordinates L*, a*, and b* of several Cr_2_O_3_ samples are much closer to those of green plant leaves when compared to undoped Cr_2_O_3_.

## Figures and Tables

**Figure 1 materials-13-01540-f001:**
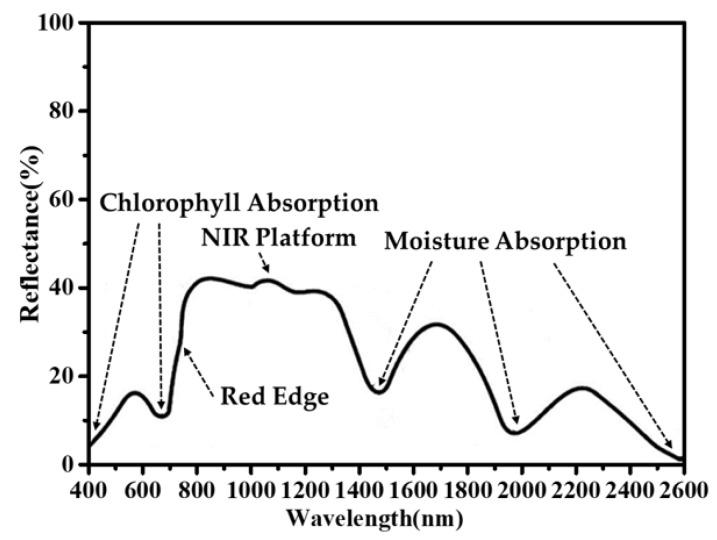
Approximate reflectance spectrum of green plants.

**Figure 2 materials-13-01540-f002:**
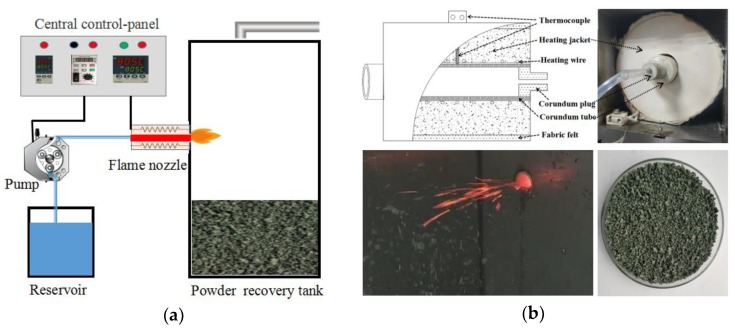
(**a**) Diagrammatic sketch of self-propagating combustion furnace. (**b**) More details about the improved solution combustion synthesis process.

**Figure 3 materials-13-01540-f003:**
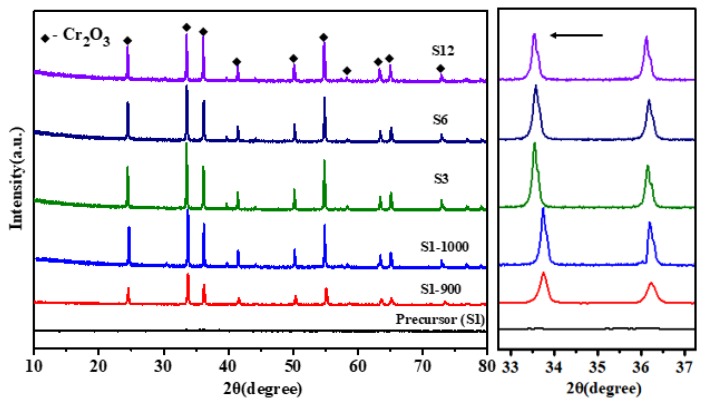
XRD patterns of Cr_2_O_3_ samples (S1-900, S1-1000, S3, S6, and S12) and the precursor (S1) with an emphasized view on the shift of the reflection peaks.

**Figure 4 materials-13-01540-f004:**
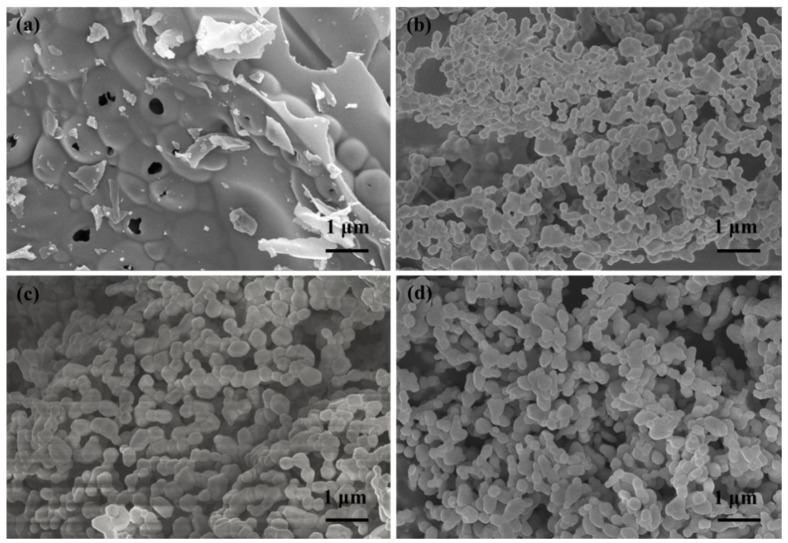
Scanning electron microscopy photographs of samples, (**a**) precursor of S1, (**b**) S1-900, (**c**) S1-1000, and (**d**) S12.

**Figure 5 materials-13-01540-f005:**
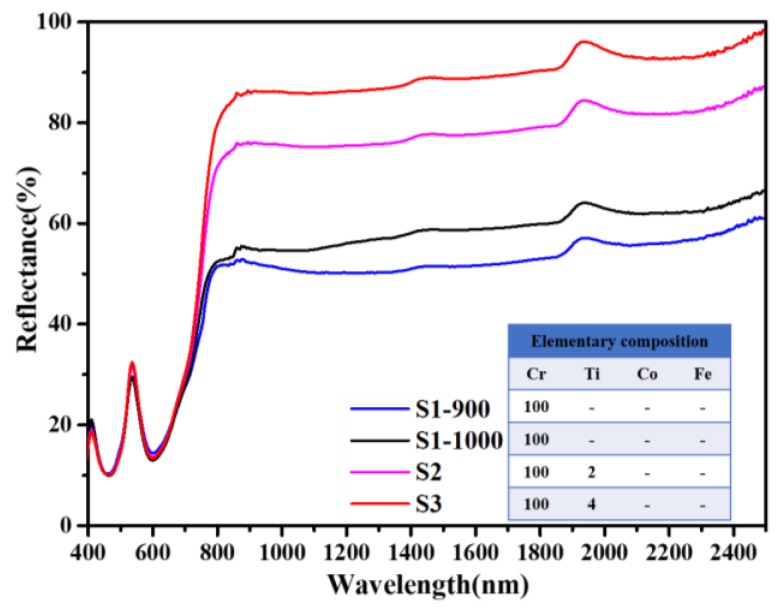
Influence of calcination temperature and Ti doping on diffuse reflectance spectra of Cr_2_O_3_ samples.

**Figure 6 materials-13-01540-f006:**
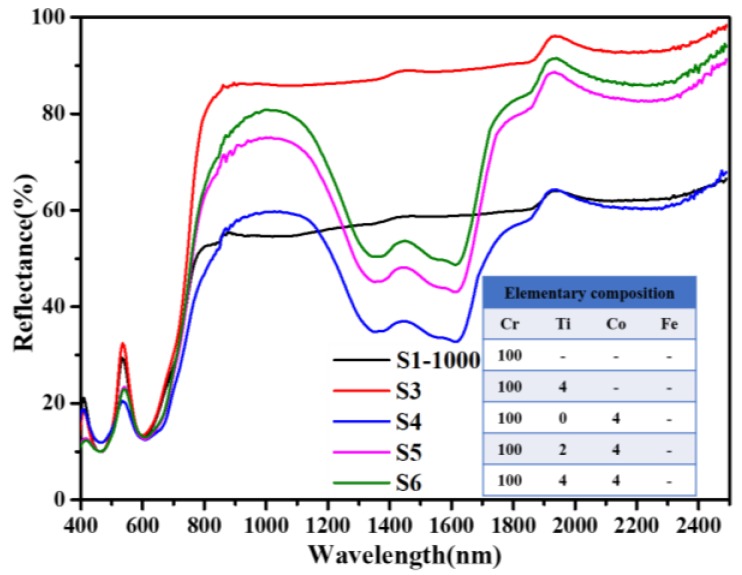
Influence of Co and Ti co-doping on diffuse reflectance spectra of Cr_2_O_3_ samples.

**Figure 7 materials-13-01540-f007:**
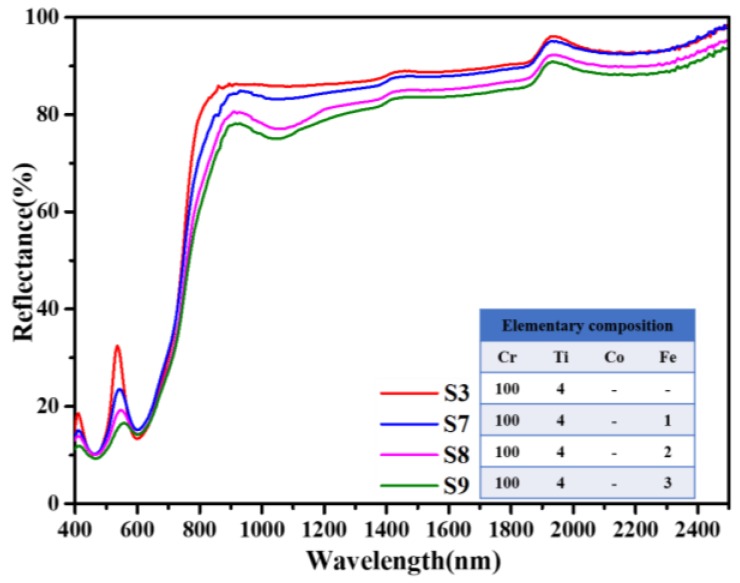
Influence of Fe and Ti co-doping on the diffuse reflectance spectra of Cr_2_O_3_ samples.

**Figure 8 materials-13-01540-f008:**
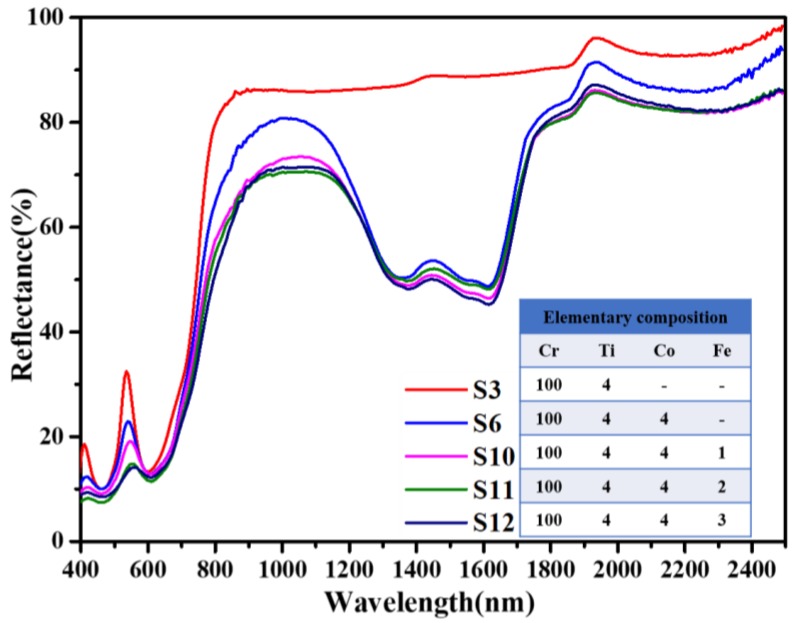
Diffuse reflectance spectra of Ti, Co, and Fe co-doped Cr_2_O_3_ samples.

**Figure 9 materials-13-01540-f009:**
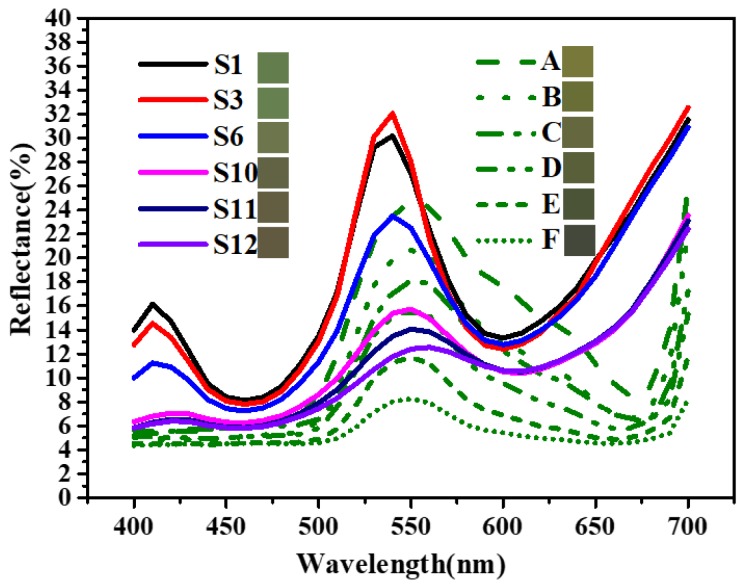
Diffuse reflectance spectra (400–700 nm) of Cr_2_O_3_ samples (S1 to S12) and Ficus microcarpa leaves (A–F).

**Figure 10 materials-13-01540-f010:**
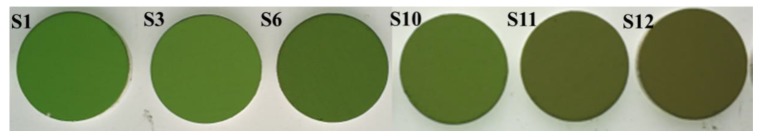
Photographs of Cr_2_O_3_ tablets prepared for the color test.

**Table 1 materials-13-01540-t001:** Elementary composition of Cr_2_O_3_ samples.

Sample	Mole Ratio				Sample	Mole Ratio			
	Cr	Ti	Co	Fe		Cr	Ti	Co	Fe
S1	100	0	−	−	S7	100	4	−	1
S2	100	2	−	−	S8	100	4	−	2
S3	100	4	−	−	S9	100	4	−	3
S4	100	0	4	−	S10	100	4	4	1
S5	100	2	4	−	S11	100	4	4	2
S6	100	4	4	−	S12	100	4	4	3

**Table 2 materials-13-01540-t002:** L*, a*, and b* values of Cr_2_O_3_ samples and Ficus microcarpa leaves.

Sample	Color Coordinates			Leaf	Color Coordinates		
	L*	a*	b*		L*	a*	b*
S1	49.61	−17.96	23.10	A	49.38	−9.14	33.63
S3	50.80	−18.22	24.45	B	44.68	−10.12	30.06
S6	47.67	−11.95	21.62	C	42.64	−7.55	23.06
S10	41.43	−6.15	17.64	D	39.53	−9.74	21.25
S11	39.81	−3.09	16.93	E	34.41	−8.41	15.22
S12	38.40	−0.72	15.58	F	30.05	−5.49	8.12

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
