# Peer review of "Preparation and Reflectance Spectrum Modulation of Cr2O3 Green Pigment by Solution Combustion Synthesis"

_materials, 2020, doi:10.3390/ma13071540_

Round 1
Reviewer 1 Report
The manuscript describes the spray pyrolysis route (if this is not spray pyrolysis the authors should explain how it’s different – calling the precursor “Amorphous” rather than a solution is misleading) to crystalline pure and doped Cr2O3 pigment. The lack of a second phase and evidence for successful doping are presented in the XRD and chemical analysis. I have no problem with the spectral analysis but would suggest the authors consider the nature of the dopant and acknowledge some previous work, especially by Atkinson et al. (Solid State Ionics 177 (2006) 1767). In their work on Ti-doping in Cr2O3, synthesized by powder routes (ball milling) they report lattice parameters for 20% and 30% Ti doping that shows a smaller down field shift (to lower two theta) that for the 4% doping level reported here. What are the reasons for this? Atkinson et al insist the doping is as Ti4+, and I find this consistent whe you look at the relative size of the possible dopants. The authors should report the hexagonal lattice parameters for their various samples.
Ion radius (Å) ion radius (Å)
Cr3+ 0.615
Ti4+ 0.605 Ti3+ 0.67
Co3+(HS) 0.61 Co2+ 0.745
Fe3+(HS) 0.645 Fe2+ 0.78
For 4% doping of Ti4+, based on Atkinson’s results, I would expect little to no change in the lattice parameters; same for Co3+. Ti4+, high spin (HS) Co3+ and Cr3+ are within 1% of the average. Also the relative shifts for samples S3, S6 and S12 are pretty much the same. One would expect no shift relative to undoped Cr2O3 for S3 and S6 and a modest downfield shift for Fe3+ (HS) doped, especially at the 4% level.
How to explain these observations? Look at the reduced dopants (Ti3+, Co2+ and Fe2+) they’re larger in ionic radius than Cr3+, on average by more than 15%. At the 4% doping level this might cause this large shift. The changes seen are probably due to reduction of some of he dopant ions. The authors should present some evidence for the oxidation state of the dopants, since the nature of the dopant clearly affects color modulation, the primary contribution of this work. Without this how can we hope to reproduce the work presented, to say nothing of understanding it. At least present a rationale of the expected oxidation states based on known redox properties of the ions (use a Pourbaix diagram and measure the fugacity or estimate it).
Reviewer 2 Report
I really do not have significant comments to this paper. This does not mean I didn't read it carefully. I did. But this paper is fundamentally correct, simple, clear and straightforward. So, I don't have any major objection. However, I think the English should be revised. Some sentences are clumsy and/or difficult to understand.
Therefore, my global appreciation is to accept with minor revisions.
Here's a list of minor comments/suggestions:
line 34: "is" not "was"
lines 34-35: this sentence is difficult to read. Please rewrite it with more care. And it seems to me that you are particularly worried about the local peak around 550 nm, but it is important to mention also the valleys caused by chlorophyll absorption at ca. 450 nm and 680 nm. You should refer to this also in the text and in the discussion
line 40: the valley around 1900 nm is never mentioned in the paper, except when is depicted in figure 1. So, please add a little bit more discussion about it
line 51: "defects" not "defect s"
line 55: "precious"? Or "previous"?
line 57: "was hardly ever" - rewrite, there is a verb missing. "Ever been"? But "was hardly ever been" it's too clumsy...Think in another expression.
line 59: persuasiveness? Are you sure this is the word you want?
line 70: The way you start section 2 gives a wrong impression to the readers. Indeed, starting with "Ti/Co/Fe doped..." suggests that only the preparation of the doped versions will be explained. So, I suggest you start this section with something like "Pure Cr2O3 and Ti/Co/Fe doped..."
line 89: "constant" is more simple and clear than "invariable"
lines 86-93: about the self-propagating combustion furnace: I think you should explain a little bit more the system, since not all the readers will have background in these type of techniques (part of the readers will have background in optics). Specifically you should explain if the combustion is triggered by the nozzle temperature only. Maybe a reference to combustion synthesis is also of help to the reader.
line 106: "The patterns of the precursor..." - no, there is no pattern at all, it's a flat line. So, you should rewrite as "the absence of patterns in the precursor..."
Figure 3: when figure 3 is shown, there is no information yet about the calcination temperature of the doped samples (this will be explained only in line 152). So, I think this information should be given already in figure 2, although the complete explanation may be given only in line 152.
line 151: "rest samples" or "remaining samples"?
Final suggestions:
1. Add information about width of the features (the width of the water 1400 nm valley vs. the width of the S4 valley at the same wavelength)
2. Should make a remark that the water absorption features at 1900 nm and 2500 nm are not yet simulated
3. On the good side, maybe you should stress that, at least in the visible part of the spectrum, there is a good match.
4. A few words about future research directions are missing.
